# A Novel Assay for Profiling GBM Cancer Model Heterogeneity and Drug Screening

**DOI:** 10.3390/cells8070702

**Published:** 2019-07-11

**Authors:** Christian T. Stackhouse, James R. Rowland, Rachael S. Shevin, Raj Singh, G. Yancey Gillespie, Christopher D. Willey

**Affiliations:** 1Department of Neurosurgery, University of Alabama at Birmingham, Birmingham, AL 35294, USA; 2Department of Radiation Oncology, University of Alabama at Birmingham, Birmingham, AL 35294, USA; 3Department of Physics, Ohio State University, Columbus, OH 43210, USA; 4Center for Clinical and Translational Science, The University of Alabama at Birmingham, AL 35294, USA; 5Institute of Regenerative Medicine, LifeNet Health, Virginia Beach, VA 23453, USA

**Keywords:** *Glioblastoma multiforme* (GBM), patient-derived xenografts (PDX), NanoString, microtumors, spheroids, heterogeneity, drug screening

## Abstract

Accurate patient-derived models of cancer are needed for profiling the disease and for testing therapeutics. These models must not only be accurate, but also suitable for high-throughput screening and analysis. Here we compare two derivative cancer models, microtumors and spheroids, to the gold standard model of patient-derived orthotopic xenografts (PDX) in *glioblastoma multiforme* (GBM). To compare these models, we constructed a custom NanoString panel of 350 genes relevant to GBM biology. This custom assay includes 16 GBM-specific gene signatures including a novel GBM subtyping signature. We profiled 11 GBM-PDX with matched orthotopic cells, derived microtumors, and derived spheroids using the custom NanoString assay. In parallel, these derivative models underwent drug sensitivity screening. We found that expression of certain genes were dependent on the cancer model while others were model-independent. These model-independent genes can be used in profiling tumor-specific biology and in gauging therapeutic response. It remains to be seen whether or not cancer model-specific genes may be directly or indirectly, through changes to tumor microenvironment, manipulated to improve the concordance of in vitro derivative models with in vivo models yielding better prediction of therapeutic response.

## 1. Introduction

*Glioblastoma multiforme* (GBM) is the most common form of primary brain cancer with a dismal median survival of approximately 18 months [1]. The current standard of care for GBM is maximal safe surgical resection followed by a 6 week course of radiotherapy (typical dose is around 60 Gy) with concomitant systemic therapy using alkylating agent temozolomide (TMZ) (75 mg/m^2^ daily), followed by 6–12 months of adjuvant TMZ (150–200 mg/m^2^ for 5 days every 28 days) [2]. Over the past decades, there have been few advancements in the treatment of GBM since the findings of Stupp et al. [2]. This may be due largely to the dearth of reliable preclinical models which accurately recapitulate the disease characteristics of GBM in patients. Currently, the best preclinical models of GBM are patient-derived xenografts (PDX). However, these models are expensive, time consuming, and not scalable for high-throughput screening of therapeutic compounds [3,4].

Traditionally, in vitro monolayer culture or GBM spheroids including immortalized glioma cells such as U87, have been used for drug candidate screening [5,6]. More recently, it has been shown that 3D cultures of GBM microtumors grown in a human extra-cellular matrix provide more reliable and relevant preclinical models for drug screening [7,8,9]. Three-dimensional (3D) microtumors are scalable for high-throughput assays while being a more reliable preclinical model than traditional spheroid culture by better recapitulating the characteristics of GBM seen in the patient. Microtumors can then be seen as an intermediate between the high complexity of a GBM-PDX model and a less complex spheroid model. Manipulations of microtumors to better recapitulate the complexity of the tumor microenvironment can increase their accuracy as a valid preclinical model [8].

Next-generation sequencing approaches can be used to profile GBM expression. However, these technologies are resource intensive and can be cost prohibitive. New, high-throughput technologies are needed for profiling GBM and gauging therapeutic response. Targeted gene expression analysis using the NanoString nCounter Analysis System from Nanostring Technologies (Seattle, WA, USA) enables the profiling of hundreds of mRNAs simultaneously with high sensitivity and precision while providing highly reproducible data over 5 logs of dynamic range [10]. Here we describe the creation of a custom NanoString assay which characterizes several canonical phenotypes of GBM including a novel molecular subtyping gene signature. We use this custom assay to profile multiple GBM-PDX lines grown as spheroids, microtumors, and PDX to evaluate the differences in gene expression among models. We also use this assay to correlate with the therapeutic response of various agents. We demonstrate that using this custom 350 gene NanoString assay, we can rapidly profile GBM samples from various models and use it in conjunction with preclinical screening of novel therapeutics.

## 2. Materials and Methods

### 2.1. Patient-Derived Xenografts

De-identified patient samples of primary GBM tissue were collected by the UAB Brain Tumor Animal Model (BTAM) Core following IRB protocol X050415007. All animal studies were approved by The University of Alabama at Birmingham Institutional Animal Care and Use Committee (IACUC−10024). Fresh tumor tissue was disaggregated and implanted in the flank of immunocompromised athymic nude (*nu/nu*) mice without intervening culture as previously described [11]. Tumors that arise are passaged from mouse to mouse without intervening passage in tissue culture. Tissues are cryopreserved from each passage to permit re-establishment of lower passage tumors.

### 2.2. Microtumor and Spheroid Culture

Microtumors were generated by embedding PDX cells into a 3D human biogel (HuBiogel, LifeNet Health, Virginia Beach, VA, USA) matrix, absent serum, as previously described [11]. Fresh PDX samples (Glioma PDX models) were procured from the UAB BTAM and tumor fragments (0.6–0.8 cm^3^) were subjected to a controlled enzyme digestion protocol (Miltenyi kits, Miltenyi Biotec Inc., Auburn, CA, USA) to produce a cell suspension. Dissociated tumor cells or spheres were maintained in defined neuro-basal media (60%–70% viability). In brief, cells (10–20 K/bead) were mixed in cold HuBiogel (3 mg/mL) and tumor beads (150–200/model) were produced via brief gelation step and cultured at 37 °C in multi-well plates.

Spheroid cultures were maintained in NeuroBasal serum-free media made using the following recipe: 500 mL NeuroBasal medium, 10 mL B-27 supplement without vitamin A, 5 mL Amphotericin (Fisher cat# MT30-003-CI, Thermo Fisher Scientific, Waltham, MA, USA), 0.5 mL Gentamycin (Fisher cat# MT-30-005-CR, Thermo Fisher Scientific), 5 mL L-glut (260 mM), 100 μL EGF (10 ng/mL), FGF-β (10 ng/mL), and 5 mL N2 supplement (Life Tech cat# 1752048, Thermo Fisher Scientific).

### 2.3. NanoString Custom Assay

PDX cells, microtumors, and spheroids were prepared by the UAB Nanostring Laboratory following standard protocols for the NanoString assay. RNA was isolated from the samples using the PureLink RNA Mini Kit (Thermo Fisher, Cat# 12183018A, Thermo Fisher Scientific) then quantified using NanoDrop. If concentration was below 20 ng/μL, RNA was concentrated using RNA Clean & Concentrator-5 (Zymo Research, Cat# R1015, Zymo Research, Irvine, CA, USA). RNA was then hybridized with the codesets and run on the NanoString Prep station before being run on the NanoString Analysis station.

The Nanostring system is attractive for preclinical and translational work due to the ease, sensitivity, reproducibility, and most importantly, customization of the chip. Various molecular signatures of phenotypic traits including, but not limited to: (a) molecular subtyping; (b) tumor microenvironment; (c) radiation resistance; and (d) stemness were curated from the literature and from sources in-house to include in our custom assay. Table 1 lists the 16 signatures included on the pan-GBM custom NanoString chip and their sources. Appendix A shows a complete listing of the genes for each of these signatures and the final build for the custom chip. The chip includes 3 gene expression molecular subtype signatures, 1 novel signature generated by methods described in this publication and 2 taken from externally published sources [12,13,14].

To test the gene signatures, GBM PDX samples were analyzed using an Affymetrix Human Exon 1.0 ST Array. The data were normalized using Robust Multichip Average (RMA) normalization performed using the Bioconductor “affy” package, log2 transformed, and scaled/centered using z-scores [15]. Probes were annotated by gene for the expression data. Duplicate genes mapping to more than one probe were collapsed into one feature and normalized for each sample by geometric mean. These data were then combined with array data from The Cancer Genome Atlas (TCGA). A novel protocol to reduce the dimensionality of the individual signatures while maintaining the predictive power of the signatures was applied in order to fit all 16 signatures onto a 350 gene chip.

The first step in the reduction was to filter genes with less predictive power from each signature by applying finite Gaussian mixture modeling to each gene. Each gene was fit to an optimal model for all samples using the “pdfCluster” function in R [16]. Genes that predicted a single cluster across all samples were deemed to lack predictive power and removed from the original signature gene list. A second reduction was then applied to the reduced signature gene list to remove genes which possessed high homology across all samples. The “caret” R package was used to generate a correlation matrix and then find all genes with Spearman correlation coefficients above 0.75 [17]. Genes which are highly correlated are deemed not to contribute significantly to the predictive power of the signature because they provide redundant information. These genes were removed from the gene filtered signature list to generate the final gene signature. Before and after reduction, Spearman correlation matrices and heat-maps were compared for each signature to confirm that the predictive power of the original signature was preserved in the final gene signature (Figure 1).

### 2.4. Novel Molecular Subtype Classification: FastEMC

We used a modified Exponential Monte Carlo (EMC) method to perform feature selection for each canonical molecular subtype (i.e., mesenchymal, neural, proneural, and classical). The standard EMC algorithm relied on a single objective function, f, that takes as input a set of features and outputs a score for the features [25]. Monte Carlo importance sampling was performed over features and simulated annealing was used to produce a feature set that optimizes f. Our modified algorithm called FastEMC is faster than EMC and offered similar performance (https://code.osu.edu/rowland.208/FastEMC). FastEMC uses multiple target functions to improve the performance of EMC. The first target function $f_{ast}$ is computationally easy to evaluate but corresponds only roughly with feature performance. The second target function $f_{ull}$ is computationally difficult to evaluate but represents the best estimate of feature performance. FastEMC uses Monte Carlo-simulated annealing with target function *f**_ast_* to find important features quickly. The features were periodically checked against target function $f_{ull}$ to be added to a list of important features.



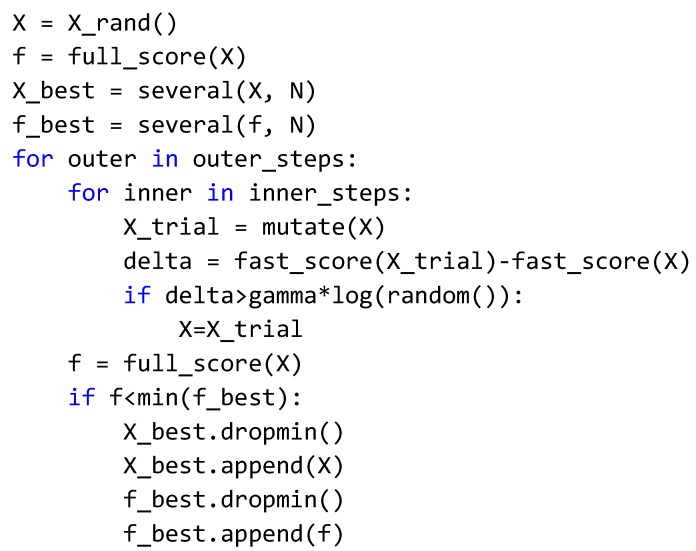



The feature set was initialized to a random selection of 20 of the possible 19,663 probes from the normalized, scaled GBM PDX data. We used classifier F-score with five-fold cross validation as our objective function. The classifier was supervised by subtype assignments from the “clanC” R program on the GBM PDX samples (25). The $f_{ast}$ function was trained with 10× fewer training iterations than $f_{ull}$. We relied on early stopping to regularize our classifier. Due to our limited number of samples we did not reserve an independent validation set to evaluate. With more samples a separate validation dataset should be used to evaluate $f_{ull}$. We compared decision trees, discriminant analysis, logistic regression, support vector machines, nearest neighbor classifiers, nearest centroid, K-nearest neighbor, k-medoids, ensemble classifiers, and neural nets. Nearest centroid, logistic regression, and support vector machines (in that order) tended to perform best, measured by F-score of the final reduced feature set. We reserved a validation set to test final performance. We used 10,000 outer iterations and 20 inner iterations. The number of feature sets in the best set list is 40. The mutation during each inner step included randomly swapping between 1 and 20 (all) of the features. We found γ=0.03 to be optimal.

The overall classifier F-score can be tested by performing unsupervised prediction of the class membership of samples for which the actual class is already known. The actual class comes from the Verhaak et al., 2010 classification method [26]. The classifier’s power in discriminating between the 4 classes can be visualized with a confusion matrix (Appendix A) or by plotting the distance (y) of each sample (x) from the centroid of each class (Appendix A). Accurate class assignments in the confusion matrix are found on the diagonal in Appendix A. In Appendix A, a smaller distance (y-axis) indicates that the sample likely belongs to that class. The confusion matrix shows that 9 samples were misclassified out of 118 total samples (92.4% combined accuracy). Appendix A is representative of one instance of the iterative classifier construction using our 100 features selected from FastEMC. The individual classifier accuracies were all in the 80%−90%+ range. Overall, the predictive accuracy combining the classifiers was ≈ 92.3%. Appendix A shows a bi-clustered heatmap of 118 GBM PDX samples using hierarchical clustering of the 100 gene features from FastEMC. Samples of the pre-determined subtypes cluster together with few exceptions and distinct patterns of gene expression can be seen for each subtype.

As an exercise, we also constructed class-specific classifiers using FastEMC supervised by the Brennan et al., 2013 published class assignments which includes the Verhaak et al., 2010 classes and an additional G-CIMP class [26,27]. When tested against a cohort of all GBM samples available from TCGA, unsupervised predictions from our class-specific classifiers had the following accuracies: “Classical” 85%, “Proneural” 86%, “Mesenchymal” 73%, “G-CIMP” 96%, “Neural” 86%.

### 2.5. NanoString Normalization

RNA isolated from GBM PDX, derived microtumors, and spheroids were run on the customized 350 gene pan-GBM chip using the nCounter Analysis System (NanoString Technologies, Seattle, WA, USA) in the UAB NanoString Core. The normalization of the NanoString data was performed using nSolver Analysis Software version 4.0 (NanoString Technologies, Seattle, WA, USA). Raw RLF files were first imported and annotations for PDX_ID and model were added. Next, background subtraction was performed using the geometric mean of 8 negative control probes. Background subtraction subtracts estimated background from the raw count. Probes less than background were assigned a value of 1. Next, a positive control normalization factors were generated using the geometric means of 6 positive control probes. Finally, a CodeSet Content normalization was performed using the geometric mean of the following reference probes: CC2D1B, GPATCH3, MRPS5, PIK3R4, and SF3A3. These genes were selected because of their low level of variance across all samples as determined by their percent coefficient of variance and from their high Spearman correlation coefficient amongst themselves as determined from previous Affymetrix data. Normalized data was then exported as a tab delimited file for further analysis.

### 2.6. Drug Screening

Microtumors (5–6 per group) and spheroids were tested for anti-tumor activity profiles of 6 selected drugs (Table 2) using EC25 or EC50 treatment doses (0–10 μM range) based on literature [28,29,30,31,32]. Cell proliferation activity of microtumors and spheroids were determined with CellTiter-Glo assay kit during the optimized 1 to 14 day culture period. Figure 1 shows the graphical outline for the cultivation of PDX, microtumors, and spheroids and parallel drug screening and NanoString assay. Drug response was measured at day 7 and day 14 following treatment. Results of drug screening are in Appendix A.

## 3. Results

### 3.1. Microtumors and Spheroids

Orthotopic PDX cells from 11 separate lines were cultivated as microtumors and spheroids for drug screening and comparative analysis. Microtumors grown in HuBiogel and spheroids grown in NeuroBasal media have distinct spheroid shapes. Calcein AM images of select microtumors are shown in Figure 2A. Comparison of immunohistochemistry (IHC) and Haemotoxylin and Eosin (H&E) staining between orthotopic PDX and derivative models show that these derivatives closely resemble the orthotopic tumor. Derivative models contain stem-like cells as evident by CD133 staining and actively proliferating cells as evident by Ki67 staining seen in Figure 2B. Gilbert et al. (2018) provides more details describing spheroids and microtumors [9].

### 3.2. NanoString Model Specific and Independent Effects

The custom NanoString panel contains 350 genes relevant to GBM biology. To judge the concordant expression of these genes between models, the directionality of gene expression in derivative models was tested for each PDX line. Z-scores were calculated for each gene using average expression across all samples. The directionality of gene expression between models were then compared pair-wise for each PDX line (i.e., orthotopic cells vs. microtumors, microtumors vs. spheroids, etc.). If the product of the pairwise comparison was positive, the genes were said to be concordant between models. If the product was negative, the genes were discordant between models. Figure 3A shows the model overlap of genes from the original 350 gene panel. There were 113 core genes found to be concordant between all three models and 85 genes showed no overlap between any of the models. The remainder of the genes showed overlap only in specific models with 25 genes appearing to be spheroid specific.

To evaluate which PDX lines had the greatest concordance between models, pairwise Pearson correlation coefficients were calculated for each PDX line compared to its derivative models. Both the entire 350 gene panel and the 113 core genes were used. As can be seen in Table 3, the correlation coefficients are stronger between models when using the 113 core genes compared to the entire panel. The 350 Gene Panel and 113 Core Genes sections are sorted in order of increasing Pearson correlation coefficients for PDX cells to microtumors. Note that the relative rankings of PDX line concordance is changed from the 350 Gene Panel to the 113 Core Genes. This further suggests that the 113 core genes are more relevant to tumor specific genes independent of derivative model. Variance filtering of the 113 core genes yielded a 17 gene signature (Figure 3B) which clustered the samples by PDX line of origin, despite the derivative model. This shows that these genes can be used to differentiate between different tumors and suggests that the 113 core genes are relevant to tumor specificity, independent of derivative model. The gaps in Table 3 are for PDX which did not have spheroids generated or tested on the NanoString assay.

All genes from the 350 gene panel were tested using the Kruskal–Wallis test (gene ~ derivative model) for variance to assess which genes were correlated with derivative model specificity. A signature of 24 genes (Figure 3C) were found to be correlated with model across all samples (*p* ≤ 0.01) and this signature clusters samples mostly by orthotopic cells versus the other two derivative models. The only two exceptions were X1465 and X1066, which are the two most concordant PDX lines between PDX cells and derived microtumors as seen in Table 3. These genes fall in the no-overlap category from Figure 3A, suggesting that these genes are derivative model specific.

### 3.3. Drug Screening

Drug screening in derivative microtumor and spheroid models was conducted in parallel to NanoString profiling. Axitinib and Erlotinib response were tested in microtumors and spheroids while TMZ response was additionally tested in orthotopic PDX. TMZ response was also measured in spheroids, however, none of the spheroids were sensitive to TMZ treatment therefore differential expression analysis in spheroid TMZ response was unable to be conducted. Differential expression analysis was performed to determine genes differentially expressed between drug responders and non-responders (*p* ≤ 0.05 and log_2_ Fold Change ≥ 2). The results of the differential expression analysis are found in Figure 4. HOXB8 was found to be over-expressed in Axitinib responding microtumors while being under-expressed in both Erlotinib responding microtumors and spheroids. These results suggest opposing roles for HOXB8 in Axitinib and Erlotinib response, and that HOXB8 downregulation is important in Erlotinib response independent of model. MGMT and AQP4 were downregulated in Axitinib responding microtumors and spheroids, respectively, but upregulated in Erlotinib responding microtumors. This suggests opposing roles for MGMT and AQP4 in Axitinib and Erlotinib response. PDGFRA was overexpressed in Axitinib responding microtumors while being downregulated in Erlotinib responding spheroids and in TMZ responding orthotopic PDX. These results suggest opposing roles for PDGFRA in Axitinib response versus TMZ and Erlotinib response. PDGFRB is over-expressed in TMZ response in both microtumors and orthotopic PDX. These results suggest the PDGFRB overexpression is relevant to TMZ response regardless of model. In PDX cells, PDGFRA is downregulated in TMZ response while PDGFRB is upregulated. These results suggest that the expression of PDGFRB favors sensitivity to TMZ while expression of PDGFRA contributes to TMZ resistance.

### 3.4. Temozolomide Concordance

Drug screening with TMZ was performed in vitro with microtumors and spheroids and in orthotopic PDX. In some of the PDX lines, derivative model TMZ response was concordant with in vivo TMZ sensitivity while it was not in others. Of the four most concordant PDX models from Table 3, X1465 and X1066 derivatives were concordant with TMZ response in vivo while X1441 and X1052 were not. The Kruskal–Wallis test (gene~TMZ response) for variance was used to assess which genes were associated with TMZ response in these four lines. A total of 38 genes (Appendix A) were found to be significantly associated with TMZ response (*p* ≤ 0.05). Differential expression analysis was performed comparing TMZ response concordant and discordant PDX lines (*p* ≤ 0.05 and log_2_ Fold Change ≥ 2) shown in Figure 5. The intersection of genes from Appendix A associated with TMZ response with the differentially expressed genes between PDX lines was observed. There were no significantly differentially expressed genes between concordant PDX cells and microtumors suggesting the closeness of these models (Figure 5A). APOD was overexpressed and CHI3L1 was under-expressed in concordant PDX lines compared to discordant PDX lines (Figure 5B–E). Generally, APOD expression is up in TMZ response concordant PDX while CHI3L1 is down (Figure 5G). This suggests that the expression of these genes may be of particular importance for model concordance with respect to TMZ response. The intersection of TMZ response associated genes with differential expression between concordant and discordant microtumors yields a signature of seven genes shown in Figure 5G which largely cluster TMZ response concordant PDX lines from discordant lines. We surmise that these seven genes (LGALS3, CHI3L1, MT1X, EPAS1, APOD, NOL4L, and MEX3B) are important for TMZ response concordance between models.

## 4. Discussion

The development of accurate, high-throughput patient-derived models of cancer is essential for profiling disease and testing novel therapeutics. Orthotopic PDX models are currently the most accurate for therapeutic testing, but these models are not well suited for high-throughput screening. Next generation sequencing (NGS) analysis is the most descriptive in profiling disease and drug response. However, NGS data is highly complex, costly, and requires extensive time and resources to analyze. Derivative models such as 3D microtumors or spheroids can be employed as a high-throughput alternative to PDX models. At the macroscopic and cellular levels, these models resemble tumors grown in vivo. A targeted, pan-GBM NanoString assay can be used to rapidly profile derivative models requiring less cost, time, and resources than NGS approaches.

The pan-GBM 350 gene NanoString assay allows for the rapid profiling of new tumors based on 16 descriptive gene signatures. Comparison of three patient-derived models reveals a set of 113 core genes which are independent of model and related directly to tumor biology. These core genes can be used to evaluate the therapeutic response of tumors. HOXB8, a homeobox DNA-binding transcription factor associated with developmental processes, is overexpressed in microtumors that are sensitive to Axitinib. However, this transcription factor is down-regulated in tumors sensitive to Erlotinib despite both Axitinib and Erlotinib being tyrosine kinase inhibitors. Erlotinib, which specifically targets EGFR, is according to our data more effective in tumors which show increased EGFR expression. MGMT expression is down-regulated in Axitinib response while MGMT is upregulated in Erlotinib response. Increased MGMT expression is already know to be associated with poor response to TMZ. Our data suggests that tumors with high MGMT expression may be responsive to combination therapy with Erlotinib. PDGFRA expression is low in Erlotinib and TMZ responders, but high in Axitinib responders. PDGFRA expression in primary tumors may then be used to predict whether a tumor will be responsive to conventional TMZ treatment, or if a combinational therapy with Axitinib may be more effective. Finally, PDGFRB expression is high in TMZ responders while PDGFRA is low. The ratio of PDGFRA to PDGFRB expression in primary tumors may be predictive of tumor response to TMZ. If PDGFRA is high in primary tumors, this may again suggest a combination therapy with Axitinib may be more effective.

Certain patient-derived tumor models may have greater concordance with drug response to orthotopic models. We have identified seven genes from our 350 gene panel which are good predictors of TMZ response concordance among models. Expression of two of these genes, APOD and CHI3L1, is consistent in comparing concordant models to discordant models. The general trend of APOD expression being up in TMZ response-concordant PDX, while CHI3L1 is down, suggests that the relative expression of these two genes may be predictive of TMZ response concordance. Future studies may find that over expression of APOD with concurrent knock-down of CHI3L1 in discordant models may improve TMZ response concordance.

We hypothesize that genes which are responsible for model concordance or which are model-dependent could be manipulated to produce more accurate derivative models of GBM. This could be accomplished through direct modulation of gene expression via over-expression constructs or selective knock-down of genes. Future studies will examine specific gene manipulation as well as focus on direct alteration of derivative model tumor microenvironment such as altering pH, glucose availability, or partial pressure of oxygen to better resemble in vivo conditions.

Utilizing our custom 350 gene NanoString panel and three patient-derived GBM cancer models, we have demonstrated that we can sort out model-dependent and model-independent effects. We can use this panel to rapidly profile new patient tumors and to potentially predict drug candidates for effective combinational therapies. We can also identify gene expression profiles which are predictive of drug response, model concordance. These findings can be utilized in future efforts to improve the concordance of patient-derived cancer models with orthotopic models to ultimately yield models more relevant to primary patient tumors.

## Figures and Tables

**Figure 1 cells-08-00702-f001:**
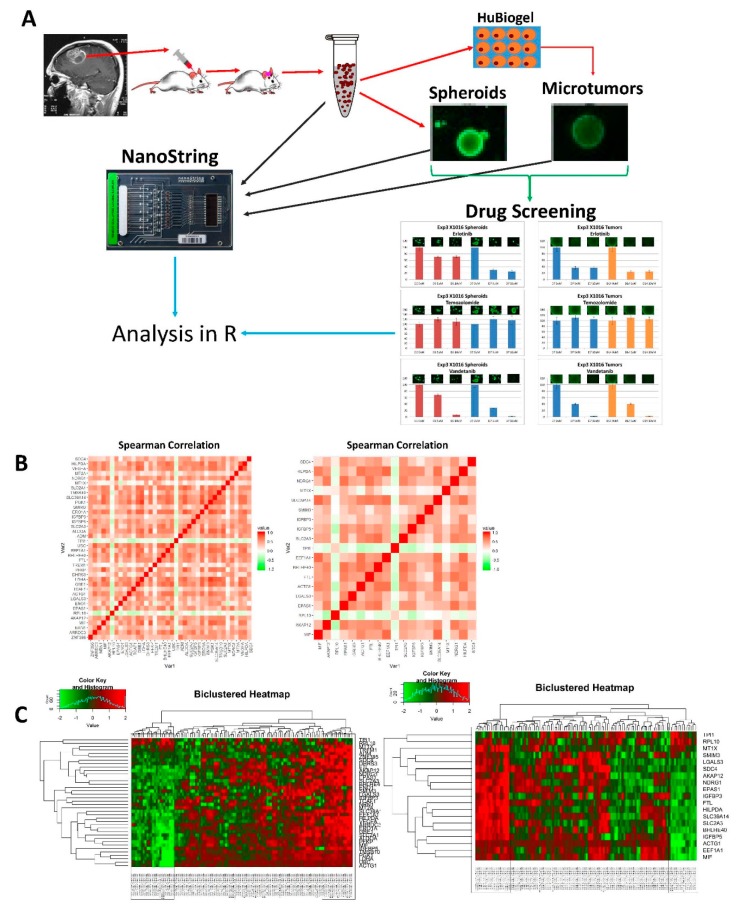
Study methods and representation of NanoString signature construction: (**A**) Graphical representation of workflow. R analysis includes differential expression, correlation, and clustering analysis; (**B**) Spearman correlation of genes in hypoxia signature before (left) and after (right) dimensional reduction; (**C**) Heatmap of samples clustered using genes from original 40 gene hypoxia signature (left) and heatmap of samples clustered using genes from dimensionally reduced 19 gene signature. Boxes highlight groups of samples with high average expression and low average expression.

**Figure 2 cells-08-00702-f002:**
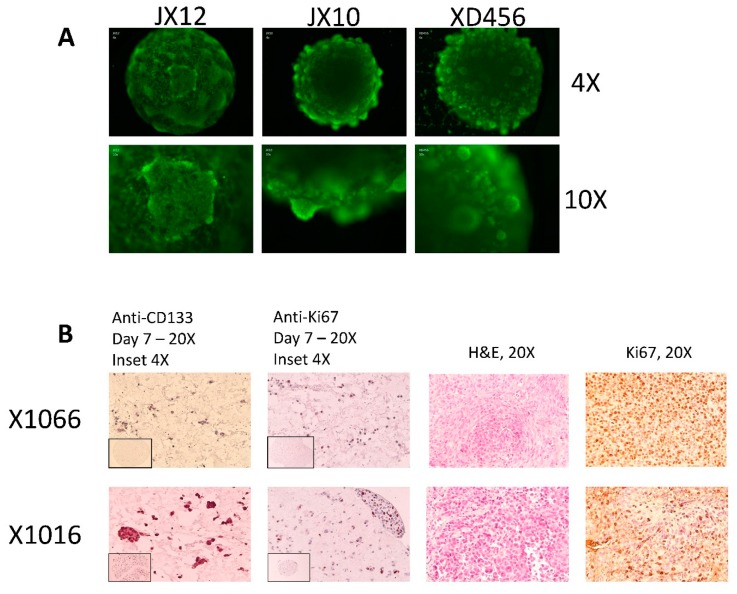
Representative images and staining of derivative microtumors: (**A**) calcein acetoxymethyl ester (AM) imaging of microtumors; (**B**) (left) IHC, anti-CD-133 staining of microtumors, (center-left) IHC, anti-Ki-67 staining of microtumors, (center-right) H&E staining of matched orthotopic patient-derived orthotopic xenografts (PDX) tumor, (right) IHC, anti-Ki-67 staining of matched orthotopic PDX tumor.

**Figure 3 cells-08-00702-f003:**
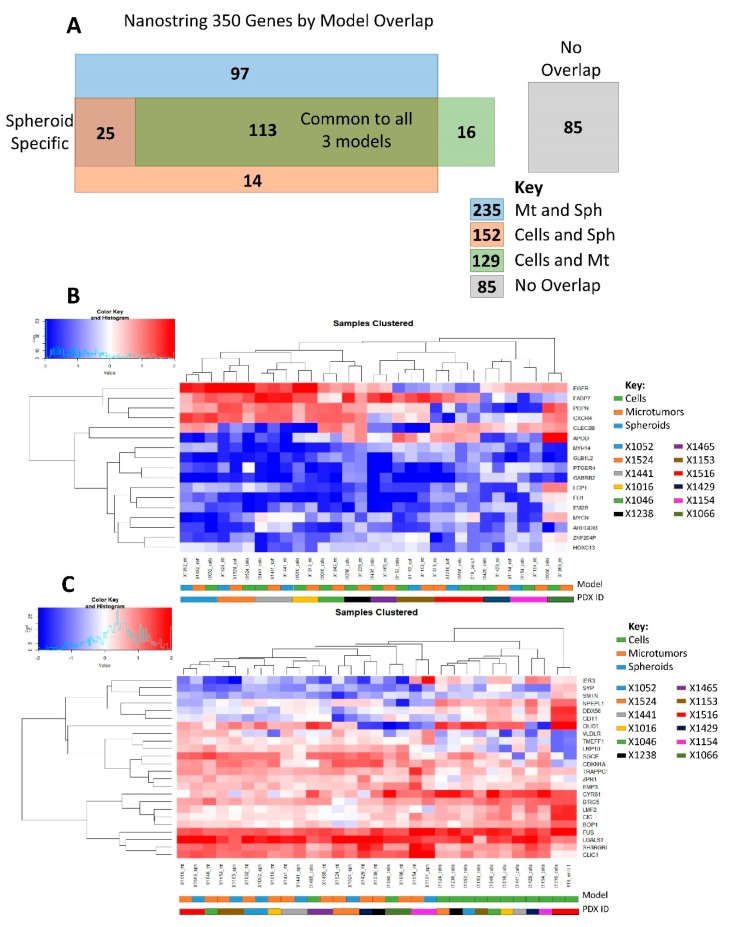
Model specific and independent effects: (**A**) Overlap of 350 NanoString genes between three different patient-derived models; (**B**) Heatmap of 17 genes derived from the core set of 113 genes which cluster samples by tumor of origin; and (**C**) Set of 24 genes found to be associated with tumor model (Kruskal–Wallis *p* ≤ 0.01) which cluster samples based on tumor model.

**Figure 4 cells-08-00702-f004:**
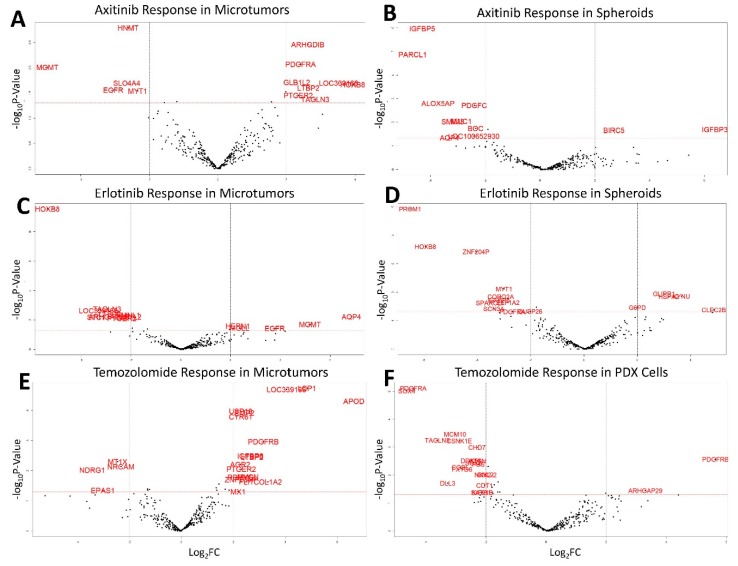
Drug screening differential expression results for Axitinib, Erlotinib, and Temozolomide: (**A**) Differential expression of genes related to Axitinib response in microtumors; (**B**) Differential expression of genes related to Axitinib response in spheroids (*n* = 6); (**C**) Differential expression of genes related to Erlotinib response in microtumors (*n* = 12); (**D**) Differential expression of genes related to Erlotinib response in spheroids (*n* = 6); (**E**) Differential expression of genes related to Temozolomide response in microtumors (*n* = 12); (**F**) Differential expression of genes related to Temozolomide response in orthotopic PDX cells (*n* = 10). Significance of red labeled genes determined by: *p* ≤ 0.05 and −2 ≥ log_2_ Fold Change ≥ 2.

**Figure 5 cells-08-00702-f005:**
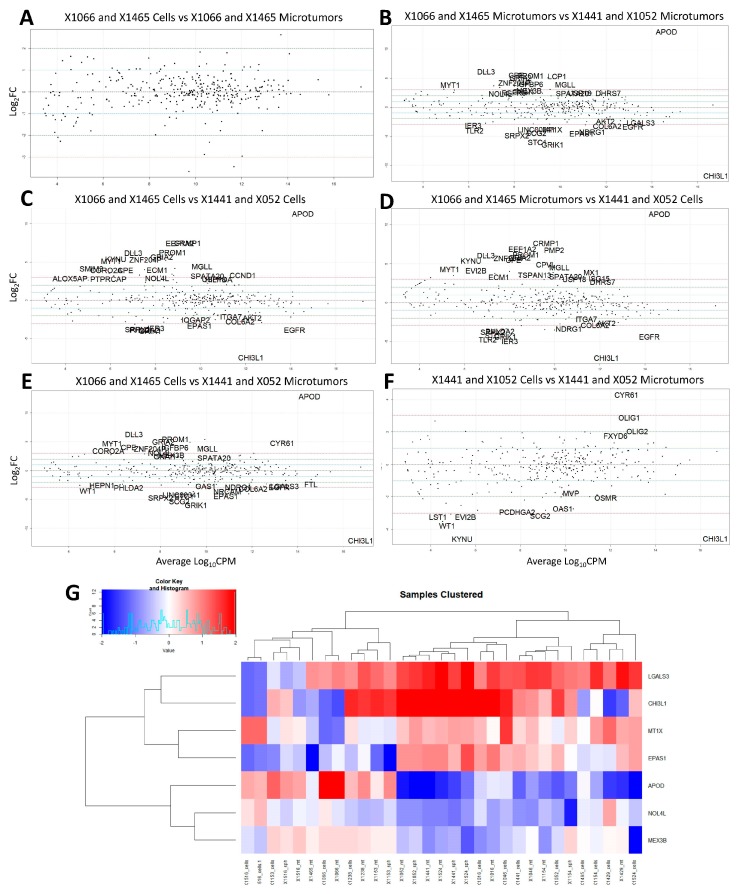
Differential expression of TMZ response concordant (X1066 and X1465) and discordant (X1441 and X1052) tumors: (**A**) X1066 and X1465 Cells vs X1066 and X1465 Microtumors; (**B**) X1066 and X1465 Microtumors vs X1441 and X1052 Microtumors; (**C**) X1066 and X1465 Cells vs X1441 and X052 Cells; (**D**) X1066 and X1465 microtumors vs X1441 and X1052 Cells; (**E**) X1066 and X1465 Cells vs X1441 and X1052 microtumors; (**F**) X1441 and X1052 Cells vs X1441 and X052 Microtumors; (**G**) Seven gene signature from differential expression of concordant microtumors vs discordant microtumors of genes associated with TMZ response. Samples clustered by TMZ response concordance (left) and discordance (right). Significance determined by: *p* ≤ 0.05 and −2 ≥ log_2_ Fold Change ≥ 2. Log_10_CPM = Log base 10 of counts per million.

**Table 1 cells-08-00702-t001:** List of gene signatures and their sources.

Signature	Genes in Signature	Source(s)
1. Novel Gene Expression Molecular Subtype	100	FastEMC
2. Gene Expression Molecular Subtype	23	Drs. Cameron Brennan and Jason Huse (Kastenhuber et al., 2014) [12,14]
3. Gene Expression Molecular Subtype	28	Patel et al., 2014 [13]
4. Cell Cycle Progression	4	Patel et al., 2014 [13]
5. Curated Genes of Interest	32	In-house
6. Genes Down-Regulated in Radiation Sensitive vs. Radiation Resistant	26	Kim, Rha et al., 2012 [18]
7. Genes Up-Regulated in Radiation Sensitive vs. Radiation Resistant	39	Kim, Rha et al., 2012 [18]
8. Positive Correlation in Radiation Resistance	37	Speers, Zhao et al., 2015 [19]
9. Negative Correlation in Radiation Resistance	37	Speers, Zhao et al., 2015 [19]
10. Radiation Sensitivity EMT Pathway	15	Meng, Fu et al., 2014 [20]
11. Hypoxia	19	Patel et al., 2014 [13]
12. Stemness	21	Patel et al., 2014 [13]
13. IFN/STAT1 Signaling	7	Willey, Gillespie et al. 2012 [21]
14. PanCancer Internal Reference Genes	7	NanoString
15. PTGER2/ptger2 Human/Mouse Reference	2	Alcoser et al., 2011 [22]
16. Other—Curated List	32	Trabelsi et al., 2016; Patel et al., 2014; Olar, Sulman et al., 2015 [13,23,24]

**Table 2 cells-08-00702-t002:** Drugs used in screening of spheroids and microtumors.

Drug	Target/Mechanism	Concentrations
Axitinib	VEGFR tyrosine kinase inhibitor	0, 5, 10 μM
Erlotinib	EGFR tyrosine kinase inhibitor	0, 5, 10 μM
Temozolomide	Alkylating agent	0, 5, 10 μM
Carboplatin	Platinum-based antineoplastic	0, 5, 10 μM
Enzastaurin	Protein kinase C beta	0, 5, 10 μM
Vandetanib	VEGFR2 tyrosine kinase inhibitor	0, 5, 10 μM

**Table 3 cells-08-00702-t003:** Pairwise Pearson correlation coefficients between models.

350 Gene Panel					113 Core Genes			
PDX_ID	Cells ^a^ to Mts ^b^	Cells to Sph ^c^	Mt to Sph		PDX_ID	Cells to Mts	Cells to Sph	Mt to Sph
X1516	0.652	0.649	0.642		X1516	0.884	0.884	0.985
X1154	0.772	0.658	0.847		X1154	0.89	0.816	0.924
X1238	0.796				X1238	0.926		
X1046	0.840				X1429	0.934		
X1429	0.845				X1524	0.941	0.942	0.962
X1524	0.857	0.874	0.943		X1046	0.954		
X1016	0.892				X1016	0.955		
X1153	0.908	0.910	0.974		X1153	0.959	0.965	0.987
X1441	0.922	0.938	0.961		X1441	0.965	0.984	0.976
X1052	0.925	0.933	0.965		X1052	0.966	0.962	0.98
X1465	0.949				X1066	0.978		
X1066	0.958				X1465	0.984		

^a^ Cells from disaggregated PDX; ^b^ Mts = microtumors generated from PDX cells; ^c^ Sph = spheroids derived from PDX cells.

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
