# Peer review of "A Novel Assay for Profiling GBM Cancer Model Heterogeneity and Drug Screening"

_cells, 2019, doi:10.3390/cells8070702_

Round 1
Reviewer 1 Report
In this manuscript the authors compare three model systems for their efficacy as a drug testing tool. For the most part the paper is written well but I have highlighted some points below which could do with some clarification. I think the novel microtumours assay described here would be a useful tool to researchers (pending some clarification).
The methodology for producing spheroids is unclear and is interspersed with the protocols describing the generation of microtumours. I would suggest these be under separate headings or at least in different paragraphs for clarity.
PDX cells are generated by disaggregation of the tissue- I would assume that this suspension also contains stromal cells? If yes then following on from this question do the beads used in the generation of microtumours select specifically for tumour cells or would stromal cells also be incorporated? What about the spheroids?
Fig 2- was the Ki67 (or any of the IHC quantitated)
Also in reference to Fig2 can the authors include images representative of spheroids for comparison?
There are multiple blank spaces in Table 3, presumably because spheroids were not generated from these samples. Can the authors clarify and include in the manuscript.
Also in relation to Table 3 Could the authors either explain why the samples are ordered as is OR re-order the PDX ID numbers so that they are in the same order for the 350 gene panel and the 113 core genes so the reader can compare directly across the row.
The authors state that X1465 and X1066 were the most concordant PDX lines between models however there no data for spheroids- they may want to adjust the statement.
Figure 4. How many samples were analysed here? Is this figure representative of one sample or a pooled group of samples? Please include an explanation and n-values.
Fig 4: were any drugs tested across all 3 models? Cells, spheroids and microtumours?
Figure 4. What is the significance of the genes highlighted in red? Why were they selected? P-value? Fold-change?
Do the authors have any data comparing their drug testing in the model systems (cells, spheroids and microtumours) to drug testing in an actual mouse with the corresponding cells injected?
Reviewer 2 Report
Stackhouse et al suggest the novel assay for profiling GBM cancer model heterogeneity and drug screening. For the analysis, NanoString method was used for targeted gene expression analysis and gene expression profiles were provided. Especially custom 350 gene NanoString panel and 3 patient-derived GBM cancer models were used. Based on this analysis, author suggest that this method can sort out model-dependent and model-independent effects. In addition, author also suggest that this method can rapidly profile the new patient tumors and this result can be used for potential prediction of drug candidates for patient treatment. The approach and analysis method are reasonable and data are clear. However, it is not clear whether the analysis method is useful for drug selection for each GBM type based on the research conclusion.
First issue: In the method part, please indicate the information about the nanostring chip (types, catalog number, company, etc.). In addition, please describe the sample preparation method.
Second issue: In this paper, 6 drugs were selected for the test. “drug screening” is “selection of drug for treatment”? In addition, in this part, drug response for the PDX cells has not been tested. Only differential gene expression profiles were provided. Author need to provide biological data.
Third issue: Have you tested the differential gene expression profile with other conventional method?
Fourth issue: In figure 1A, “R” is just flat form. Please indicate actual analysis method.
Round 2
Reviewer 2 Report
Most issues are clarified.